# DrugDevCovid19: An Atlas of Anti-COVID-19 Compounds Derived by Computer-Aided Drug Design

**DOI:** 10.3390/molecules27030683

**Published:** 2022-01-21

**Authors:** Yang Liu, Jianhong Gan, Rongqi Wang, Xiaocong Yang, Zhixiong Xiao, Yang Cao

**Affiliations:** 1Center of Growth, Metabolism and Aging, Key Laboratory of Bio-Resource and Eco-Environment of Ministry of Education, College of Life Sciences, Sichuan University, Chengdu 610000, China; snoopy_ly@yeah.net (Y.L.); gjianhong@foxmail.com (J.G.); w15645095628@163.com (R.W.); 2019222040074@stu.scu.edu.cn (X.Y.); jimzx@scu.edu.cn (Z.X.); 2Department of Data Science, Dana-Farber Cancer Institute, Boston, MA 02110, USA

**Keywords:** COVID-19, SARS-CoV-2, computer-aided drug design, bioinformatics, database

## Abstract

Since the outbreak of SARS-CoV-2, numerous compounds against COVID-19 have been derived by computer-aided drug design (CADD) studies. They are valuable resources for the development of COVID-19 therapeutics. In this work, we reviewed these studies and analyzed 779 compounds against 16 target proteins from 181 CADD publications. We performed unified docking simulations and neck-to-neck comparison with the solved co-crystal structures. We computed their chemical features and classified these compounds, aiming to provide insights for subsequent drug design. Through detailed analyses, we recommended a batch of compounds that are worth further study. Moreover, we organized all the abundant data and constructed a freely available database, DrugDevCovid19, to facilitate the development of COVID-19 therapeutics.

## 1. Introduction

On 31 December 2019, a previously unknown coronavirus was reported in Wuhan (China) to the World Health Organization. It is now known as severe acute respiratory syndrome coronavirus 2 (SARS-CoV-2), which can manifest itself in the disease named COVID-19. At the time of writing, nearly 239 million cases have been reported worldwide, with over 4 million deaths. SARS-CoV-2 has a genome of approximately 30,000 nucleotides, which can be translated into approximately 30 proteins [1,2]. The encoded spike (S) protein employs the host’s angiotensin-converting enzyme 2 (ACE2) and extracellular protease transmembrane protease serine 2 (TMPRSS2) for S attachment and priming, respectively [3,4,5]. After that, it starts the fusion of the virus envelope and cell membrane. Researchers have now revealed that the virus can enter host cells through endocytosis, and use the pH-sensitive endosomal cysteine proteases cathepsin B and L (CatB/L) for S protein priming in TMPRSS2-cell lines [4,6]. Recently, it was found that another entry cofactors called neuropilin-1 (NRP1) on the host cell surface can bind to the S protein to potentiates infectivity [7,8]. After the virus invades the cell and uncoating, the viral genome RNA is translated to produce several viral non-structural proteins, including RNA-dependent RNA polymerase (RdRp), the papain-like protease (PLpro), and 3-chymotrypsin-like main protease (Mpro). The two proteases are assembled into viral replicase–transcriptase complex [9,10].

These intensive studies reveal many attractive targets for anti-SARS-CoV-2 drug discovery and repurposing, including the viral non-structural proteins (nsp1–16), viral structural proteins (e.g., S and nucleoprotein), and host receptors and enzymes related to virus infection or reproduction in host cells, such as ACE2, TMPRSS2, DHODH, and PAR-1 [11,12]. According to the solved structures in the PDB [13], 82% (18/22) of the potential targets currently have complete or partial structural data, and the mechanism of inhibiting targets has been achieved intuitively and profoundly by the co-crystallization of targets and their potential inhibitors. It provides a strong foundation for the development of anti-SARS-CoV-2 drugs, taking advantages of CADD and high-throughput screening.

So far, a large number of drug candidates have been developed for COVID-19 treatment, some of which are tested clinically and have shown preliminary therapeutic effects [14,15,16,17,18,19,20,21]. We believe that it is the right time to summarize the discovery of these anti-COVID-19 drug candidates for a better understanding of the strong and weak points of current CADD studies. In this work, we collected 779 drug candidates discovered by CADD methods from 181 publications and conducted comprehensive comparisons on their targets, binding conformation, druggable properties, etc. We also built a publicly available database to illustrate those discoveries, which may profit the drug development against COVID-19.

## 2. Methods

### 2.1. Compound Collection and Preprocessing

The compounds in this work were collected from 181 peer-reviewed articles since the outbreak of SARS-CoV-2. The articles were obtained by searching Google Scholar and PubMed using the keywords of “SARS-CoV-2” or “COVID-19” as well as “CADD”, “Virtual Screening”, “Drug Design”, “Repurposing”, “Screening”, “Drug Discovering”, “Inhibitor”, etc. The majority of the 2D structures were downloaded from PubChem [22] or ZINC [23] databases. The compounds that were not included in either database were manually constructed according to the structure diagram provided in the literature. Their 3D structures were generated using the Experimental-Torsion Basic Knowledge Distance Geometry (ETKDG) method embedded in RDKit 2021.03.01 [24]. To conduct the molecular docking calculation based on AutoDock Vina (Version 1.1.2) [25] and CB-Dock (version 1.0) [26], the AutoDockTools [27] software was utilized to convert these compounds into the PDBQT format.

### 2.2. Protein Preparation for Molecular Docking

Among the 16 target proteins from 181 CADD publications, 15 were available in the PDB database [28]. Since most targets have multiple entries, they were selected by the following criteria: (1) structures with full-length or at least complete active sites; (2) structures with co-crystallized ligand; (3) structures used in the reported virtual screening research. Accordingly, the selected crystal structures include 6W6Y (Mac1), 6W9C (PL-PRO S1/S2 pocket), 7CMD (PL-PRO S3/S4 pocket), 6LU7 (Mpro), 6W4B (nsp9), 7BV2 (RdRp), 6XEZ (nsp13), 6W01 (nsp15), 6W4H (nsp16), 6M0J (S), 6M3M (Nucleoprotein), 1R4L (ACE2 catalytic pocket), 6M0J (ACE2 PPI surface), 7MEQ (TMPRSS2), 7K2U (DHODH), and 3VW7 (PAR-1). Particularly, due to the lack of active sites in the experimental structures of nsp1 and nsp14, we obtained their structure models from (https://zhanglab.ccmb.med.umich.edu/COVID-19/, accessed on 31 January 2020) [29] for active-site analysis and docking. For each target, water molecules and ligands were deleted, while hydrogens were added with appropriate protonation states using AmberTools [30]. Proteins with missing loops and side chains were fixed using DeepView v4.1 [31]. The binding sites of each compound were obtained by referring to the corresponding publications. In addition, we obtained the human homologous protein corresponding to each target for off-target analysis by protein–protein blast. For the 12 target proteins from SARS-CoV-2, only the Mac1 of nsp3 possessed several homologous proteins from humans, including ADP-ribose glycohydrolase MACROD1, protein mono-ADP-ribosyltransferase PARP9 (PARP9), and protein mono-ADP-ribosyltransferase PARP14 (PARP14) (with 28%, 31% and 32% sequence identity, respectively). For the four target proteins from humans, ACE2 has high homology with ACE, with 42% sequence identity. TMPRSS2 and PAR-1 have numerous homologous proteins, and according to the sequence identity, we selected the top 20 homologous proteins for each target to perform off-target analysis.

### 2.3. Docking Protocol

To investigate the binding capability of the reported compounds, AutoDock Vina 1.1.2 was employed for the docking simulation. Moreover, template-based docking simulations were also performed to illustrate the similarities and differences between them. The latter was carried out using a template-based docking tool (FitDock, in-house developed software, unpublished). For each compound, the most similar active inhibitor and its corresponding co-crystallized target structure were used as a template to perform the FitDock simulation.

### 2.4. Structure Analysis of Compounds

The chemical spatial features of CADD recommended compounds and known inhibitors, including cLogP (calculated lipophilicity), HBA (hydrogen-bond acceptor), HBD (hydrogen-bond donor), MW (molecular weight), and NRotB (number of rotatable bonds), were evaluated by XLOGP3 [32]. We evaluated the chemical space coverage by computing t-Distributed Stochastic Neighbor Embedding (tSNE) maps of chemical spatial features of the CADD compounds and known inhibitors [33]. Exploiting this method, compounds with similar chemical features can be clustered together. The structural similarity between compounds was evaluated with the Tanimoto similarity coefficient of Morgan fingerprint generated by RDKit, the similarity score of LigMate [34] and the PC-score of LS-align [35].

### 2.5. The Web Server of DrugDevCovid19

The DrugDevCovid19 web server was implemented in HTML5, CSS3, PHP, and JavaScript. The chart.js was used to visualize the statistical data and the ngl.js [36] was used to visualize the structural data which were collected in this work. The server offered chemical structures and identifiers (PubChem CID or ZINC ID or Custom ID) of compounds, structures, and PDB codes of the corresponding targets, their 3D binding conformations, sources et al. If a compound was reported multiple times, they were merged into one item. All those data were well-organized for comparison studies.

## 3. Results

### 3.1. Molecular Docking Is the Most Popular CADD Method against COVID-19

CADD has seen broad application in drug discovery against COVID-19. In our collected publications, most of them employed docking screen [37,38,39,40,41,42,43], molecular simulation [44], pharmacophore models [45,46], or machine learning-based virtual screens [47] for drug discovery. Docking screen was the most popular one, which took over 96% of the works. They took the advantage of well-established docking tools including the academic software AutoDock [27] or AutoDock Vina [25], DOCK6 [48], LeDock [49], rDock [50], LigandFit [51], etc., or the commercial software Glide [52], GOLD [53], Surflex-Dock [54], MOE [55], Discovery Studio [56], etc. A typical docking screening conducted by those studies includes three steps: (1) molecular docking, to predict binding modes and affinities of each molecule in a compound library; (2) analyzing ADMET to predict the pharmacokinetic properties; (3) molecular dynamics, to examine the binding stability and free energy. Besides the docking methods, compound libraries were also a key factor for virtual screening. They mainly included FDA-approved, investigational, and experimental drug libraries, natural product libraries, and comprehensive compound libraries, e.g., ZINC [23], ChEMBL [57], PubChem [22]. Particularly, almost one-third of our collections employed the FDA-approved, investigational, or experimental drug libraries, indicating the importance of drug repurposing in dealing with this pandemic.

A successful CADD study not only depends on computational methods but also relies on the target proteins. In our collected publications, we observed 16 drug targets used for drug design (Table 1 and Appendix A). They included the 12 proteins from SARS-CoV-2 but also 4 proteins from human, a host of the virus. The former included the viral non-structural proteins, such as nsp1~16 involved in the viral pathogenicity, replication, modification of viral RNA, or assembly of virions, and structural proteins, such as spike protein (S) related to the viral infection, or nucleoprotein (N) related to viral assembly. The latter included 4 enzymes including angiotensin-converting enzyme 2 (ACE2), transmembrane protease serine 2 (TMPRSS2), dihydroorotate dehydrogenase (DHODH) and proteinase-activated receptor 1(PARP-1). In all of those 16 proteins, 15 had at least one experimentally solved structure, except nsp14. Particularly, as of 4 August 2021, the number of experimental structures for nsp3 (PLpro), nsp5 (Mpro), and S were as many as 308, 369, and 466, respectively in the PDB [28], which are remarkably more than those for the other targets (Table 1 and Appendix A). Comparative analysis showed that some of the proteins showed significant conformational changes upon ligand binding. For example, the PLpro-inhibitor complex and PLpro apo structure were remarkably different at BL2 loop (Appendix A), and open and closed conformations were observed in the exterior residues (e.g., Q24, D30, E35, E37, D38, Y41, Q42, N53, E56, Y83, Q325, E329, Q330, K353, R393) of ACE2 (Appendix A). Those structural variations have been carefully investigated in some of the CADD studies [58,59].

### 3.2. CADD Studies Predicted Diverse Compounds against COVID-19

Since the outbreak of SARS-CoV-2, CADD has been widely used either by drug repurposing or screening novel compounds. In the 181 research reports collected, we obtained a total of 1073 compound records, which included molecular sources, research methods, targets, active sites for binding, etc. Among these compounds, 53.6%, 11.4%, 7.5%, 7.1%, 6.6% target Mpro, S, RdRp, PLpro, and ACE2, respectively (Figure 1a). Some of the compounds have been recommended by multiple research teams. For example, lopinavir, one of the compounds recommended by nine virtual screening researches, has been proved to inhibit SARS-CoV-2 with the IC50 of 9.12 μM in the later wet-lab research [60]. Thus, we summarized the compounds recommended more than twice in targeting a receptor, as shown in Figure 1b, suggesting their potential in the following research.

After eliminating the redundant records, we finally obtained 779 recommended compounds, including 457 (59%), 114 (14.6%), 74 (9.5%), 70 (9%), and 69 (8.9%) compounds targeting Mpro, S, PLpro, RdRp, ACE2, and the others, respectively (Table 2). We also observed that a target protein may contain two or more binding pockets for drug design. These targets include nsp1, PLpro, nsp9, nsp13, nsp14, nsp16, S protein, N protein, and ACE2 (Appendix A). The different pockets usually play different roles in a target protein. For example, in the papain-like protease domain of PLpro, the S1/S2 pocket is the binding regions of catalytic sites while S3/S4 pocket (the BL2 loop) is substrate-binding related sites [58], and in nsp13, both the ATP binding pocket and RNA binding pocket are used for screening [61]. In this work, we did not distinguish the detailed impact of the binding pockets, but performed the analysis by comparing the recommended compounds using the same binding pockets but not the same target proteins.

In order to compare compounds obtained from different studies, we performed unified molecular docking against the top five popular drug targets, including 457 recommended compounds against the catalytic site of Mpro, 73 recommended compounds against the PPI surface of S, 68 recommended compounds against the catalytic site of RdRp, 68 recommended compounds against S3/S4 pocket of PLpro, and 41 recommended compounds against the catalytic site of ACE2 (Figure 2a). The unified docking was compared to the nine experimentally validated compounds against Mpro, with the corresponding docking scores of ~−7.0 kcal/mol (Appendix A). We observed that 76%, 75%, 47%, and 59% recommended compounds in Mpro, S3/S4 pocket of PLpro, RdRp, and S respectively achieved lower docking scores (lower than −7.0 kcal/mol), which suggest stronger binding. Particularly, GRL0617, the ranking-first compounds targeting PLpro S3/S4 pocket (docking score = −10.0 kcal/mol) was confirmed to have strong affinity (KD = 1.93 μM) [58], after predicting in July 2020 [37]. Unlike the four targets, only 2% recommended compounds against ACE2 achieved scores were lower than −7.0 kcal/mol. It could be attributed to the flat interface between ACE2 and compounds, which originally binds with the viral S protein. Hence, it is much more challenging to develop drugs against ACE2 than the others.

We also evaluated the drug-likeness of the recommended compounds using Lipinski’s “Rule of Five” [62]. The results showed that 67% and 81% of compounds targeting Mpro and PLpro, respectively, met at least four of the Lipinski’s rules (Figure 2b). However, only 53%, 56%, and 57% of compounds targeting S, ACE2, and RdRp, respectively, were able to fulfill the criteria, implying the difficulties to discover qualified drugs against those targets.

### 3.3. Comparison Studies between CADD Compounds and Co-Crystallized Inhibitors

So far, a large number of co-crystal structures of SARS-CoV-2 targets in complex with inhibitors are available in PDB, for example, 136, 74, 37, and 16 compounds are co-crystallized with Mpro, DHODH, ATP binding domain of nsp13, and papain-like protease domain of PLpro, respectively (Appendix A). It is a valuable source for evaluating and improving the CADD recommended compounds by comparing with those experimentally determined inhibitors. In the following sections, we elaborate on the results of our comparison studies on molecular similarity, drug-likeness, and binding modes for those compounds.

#### 3.3.1. Molecular Similarity

We first analyzed molecular similarity between the compounds discovered by CADD and the co-crystallized inhibitors in terms of 2D (Morgan fingerprint) and 3D (LS-align) similarities, which were evaluated by Tanimoto coefficients and PC-score respectively. The result showed that in the available inhibitors against 11 SARS-CoV-2 targets, the average Tanimoto coefficients were all less than 0.3, indicating that the majority of CADD compounds were dissimilar to the co-crystallized inhibitors (Figure 3a). In other words, CADD compounds showed diverse chemical scaffolds compared with the co-crystallized inhibitors. Nevertheless, we also found that 19, a small number of CADD compounds, were highly similar to the Mpro inhibitors (Tanimoto similarity coefficient ≥ 0.5) (Appendix A). Among them, 14 flavonoid analogues shared the same skeleton as the known inhibitors Myricetin [63], 7-O-methyl-myricetin [63], 7-O-methyl-dihydromyricetin [63], and Baicalein [64]. In addition, perampanel is the lead compound of the known inhibitor COMPOUND4 [65], which also has the same skeleton structure. Flovagatran, DB04692, compound 23727975, and caspase-1 inhibitor VI are peptoid compounds, which have the most similar topological structures with the known inhibitor MPI1 [66], MPI6 [66], N3 [67], and Z-VAD(OMe)-FMK [68], respectively. Compared to the Tanimoto coefficients, the 3D similarity analysis showed higher similarities that the average PC-scores are all larger than 0.5. They were even over 0.7 for the compounds targeting Mpro, nsp13, and DHODH, which indicated that the compounds shared rather similar topology in space despite the different chemical composition (Figure 3b). What is more, 1.5%, 42.4%, 5.9%, 80%, 2.4%, 6.9%, and 11.1% CADD compounds targeting PLpro (S3/S4 pocket), Mpro, RdRp, nsp13 (ATP binding site), ACE2 (catalytic pocket), TMPRSS2, and DHODH, respectively, showed that maximal PC-score were greater than 0.8. The above results illustrated that the CADD methods not only potentially discovered the variants of co-crystallized inhibitors but also created diversities for further drug discovery.

#### 3.3.2. Drug-Likeness

We further performed the comparative study on the drug-likeness of the CADD compounds with the known co-crystallized inhibitors against the corresponding targets in terms of cLogP (calculated lipophilicity), HBA (hydrogen-bond acceptor), HBD (hydrogen-bond donor), and MW (molecular weight). In this section, we investigated the compounds targeting Mpro (136 inhibitors, 457 CADD compounds), PLpro S3/S4 pocket (13 inhibitors, 68 CADD compounds), and DHODH (74 inhibitors, 27 CADD compounds) because they have multiple co-crystallized structures, making the comparative analysis more compelling. According to Lipinski’s “Rule of Five”, the compounds with MW ≤ 500, HBA ≤ 10, HBD ≤ 5, and −2 ≤ cLogP ≤ 5 are predictive of good permeability and absorption [62]. Because the latest evaluation of 204 small molecule oral drugs by Brown et al. [69] showed 30% approved drugs with an MW higher than 500 Da. Therefore, we raised the MW standard in “Rule of Five” to 600 Da in the following analysis. The chemical space comparison between SARS-Cov-2 Mpro inhibitors and CADD compounds was illustrated in a tSNE map (Figure 4). Fifty-four percent of the CADD compounds met the “Rule of Five”. Moreover, the distribution of drug-likeness features was basically the same as that of Mpro inhibitors. The clusters of inhibitors showed that the peptoid inhibitors represented by PF-00835231 [70], MPI7 [66], and Narlaprevir [71] accounted for the majority (52%) of the currently known Mpro inhibitors. These inhibitors typically had greater MW and binding affinities, which was closely related to the larger binding pocket of the active site in Mpro [65,70,72]. More significantly, a cluster of CADD compounds presented in the region at which peptoid inhibitors was distributed, implying their similar drug-likeness properties. The remaining 46% of CADD compounds violated one or more rules in the “Rule of Five”. Some of them showed higher hydrophilicity, and the others had larger molecular weights (MW>600) or higher lipophilicity (cLogP>5). For PLpro and DHODH (Appendix A), 78% of CADD compounds targeting the PLpro S3/S4 pocket met the “Rule of Five” and had a more diverse chemical space than the known inhibitors. All the CADD compounds of DHODH met the “Rule of Five” and fell within the chemical space of known inhibitors.

#### 3.3.3. Binding Modes

To reveal the molecular mechanism, we next compared the binding modes of the CADD compounds with the co-crystallized inhibitors. Our analysis was focused on the 19 CADD recommended compounds of Mpro described above, which shared highly similar topological structures with co-crystallized inhibitors. We compared the docking conformation of these CADD compounds to the corresponding co-crystallized structures (Appendix A) in terms of the chemical compositions for favorable binding. Among the 19 recommended compounds, perampanel, an anti-epileptic drug, possesses a distinctive topology. Compared with the co-crystallized inhibitor (named COMPOUND4 in Zhang’s report [65]), perampanel lacked the polar interaction with His163 and Glu166 and hydrophobic interaction with the S2 pocket in its predicted binding mode (Figure 5a). Moreover, the binding assays also showed that the activity of perampanel was much lower than that of COMPOUND4 (e.g., IC50 of 100~250 μM vs. 4.02 μM [65]). Fourteen flavonoid analogues shared the similar binding mode and key interactions with the co-crystallized inhibitors. For example, the parent nucleus of Baicalin docked into the Mpro pocket in a similar manner to a co-crystallized inhibitor Baicalein, which possessed approximately 7-fold higher affinity than Baicalin (e.g., IC50 of 0.94 vs. 6.41 μM [64]). Since the 7-O-glucuronide was unable to fit in the S1’ pocket appropriately in this binding mode, it resulted in the weaker polar interactions with Leu141, Gly143, and Glu166. However, the interaction of 7-O-glucuronide with Thr24–26 on the outside of the S1’ pocket contributed to the binding of Baicalin (Figure 5b). As another example, quercetin and robinetin shared the similar binding mode with the covalent inhibitor Myricetin (IC50 of 0.63 μM [63]), and both could form polar interactions with GLU166 (Figure 5c,d). Additionally, since the pyrogallol group in Myricetin worked as an electrophile to covalently modify the catalytic cysteine of Mpro [63], it can be speculated that the pyrocatechol group in quercetin and the pyrogallol group in robinetin may also have the ability to covalently bind to catalytic cysteine. In-depth study of these flavonoids will further promote the design of non-peptidomimetic covalent or non-covalent inhibitors against Mpro. The remaining four compounds were peptoids. Among them, DB04692 and compound 23727975 shared the similar non-covalent docking modes, with identical or similar groups binding in the S1 and S2 pocket as that of co-crystallized inhibitors (Figure 5e,f). Moreover, both had a reactive warhead in the S1’ pocket, close to Cys145, implying the ability to form covalent bond.

### 3.4. An Atlas of the Inhibitors against SARS-CoV-2

To facilitate the analysis of the discovered drug candidates against SARS-CoV-2, we constructed an online database, named DrugDevCovid19, which collected the compound structures, target protein structures, chemical properties, screening methods, experimental data, and 3D docking modes from the primary published computational and experimental reports. The current version contains 779 compounds and 16 drug targets extracted from 181 reports mentioned above. It is publicly available via a user-friendly interface at http://clab.labshare.cn/covid/php/index.php (accessed on 13 November 2020) (Figure 6). Compounds and targets of interest can be searched in a variety of ways at the above webpage. The ‘Entries List’ presents all the compound records retrieved from literature. The records were classified by the drug target and displayed in a donut chart. Users can click the regions in the chart to view the detailed records. The ‘Candidates List’ presents all the non-redundant compounds contained in the records. Similarly, users can click each region to view the compounds which are sorted by record number. Since there may be multiple druggable sites on one target, the compounds were further classified according to the active site and sorted by the binding affinity values predicted by AutoDock Vina [27]. The ‘Targets List’ presents all the potential drug targets. For each target, the web interface illustrates the full name, PDB code(s), the potential active sites retrieved from literature, the organisms, and the UniProt ID. Users can click the target name to view more information, such as the target function retrieved from UniProt and the surface residues in the active site. The 3D structural features of the active site can be viewed via an NGL Viewer. For each CADD compound, we showed its chemical spatial features, drug function (retrieved from DrugBank if it exists), structural information (including SMILES, 2D, and 3D structures), and Vina scores, with an interactive 3D visualization of the docking conformation.

## 4. Discussion

To develop therapeutic drugs that specifically target SARS-CoV-2 as soon as possible, scientists have comprehensively studied the SARS-CoV-2 from various aspects, including the viral genome, proteome, the pathway of viral infection, and patient’s immune response to the virus. Among them, the proteins encoded by the viral genome and the host proteins that are involved in viral invasion are the most promising anti-SARS-CoV-2 targets. CADD can be used to quickly screen potential compounds to target these proteins. In this work, we collected 779 CADD recommended compounds against 16 targets reported in 181 CADD studies. We analyzed the structures in which the inhibitors are co-crystallized with their target proteins, explored their binding modes and conformational diversities, aiming at gaining structural insights for CADD and identifying features and rules for further drug design and refinement. Our efforts can be briefly summarized into three aspects. First, we analyzed the structures of the targets in complex with active inhibitors. We preliminarily analyzed the structural diversity, binding modes, and the structure-activity relationship of the target proteins in complex with the active inhibitors. Second, we performed docking analyses of the CADD compounds. We identified the most similar active inhibitors of each CADD compound and performed molecular docking to compare their binding modes and affinities with those of the known inhibitors to select the most promising drug candidates. Finally, we created the freely available database to visualize the data generated in the above two aspects.

Among the 779 CADD recommended compounds, 347 compounds were derived from the DrugBank database, and these compounds are known to have relatively well-defined targets or activities with a high off-target risk. For the other 432 candidate compounds outside the DrugBank database, we analyzed their potential off-target effects by comparing their Tanimoto similarity with the drugs in DrugBank, and the results showed that 7.3% of the 432 candidate compounds have a maximum similarity greater than 0.8 to the drugs in DrugBank, implies potential off-target effects. Users can explore the potential targets of the candidate compounds by ligand similarity-based methods during further screening and analyze the binding ability of the candidate compounds to the potential targets by docking. In addition, we also analyzed the human homologs of each potential target by protein–protein blast, and the results showed that there are several human homologs for the four targets: Mac1, ACE2, TMPRSS2, and PAR-1, for which the candidate compounds (47 in total) need to be noted for their off-target risk.

Among the 16 protein targets, the viral Mpro and PLpro are the most widely studied. Since the two proteases of SARS-CoV-2 are highly similar to those of SARS-CoV (96% and 83% of sequence identity for Mpro and PLpro, respectively) [73], the peptoid compounds designed in previous studies which can covalently bind to SARS-CoV Mpro as well as other proteases, were first attempted to treat SARS-CoV-2 infection [70,74,75]. Most of the Mpro inhibitors currently under investigation are peptide-like compounds with a reactive head that can covalently bind Cys145. VIR250 and VIR251, two PLpro inhibitors developed by Rut et al., are also peptoid molecules that can covalently bind to the catalytic residue Cys111 [76]. However, such compounds may be ineffective due to the degradation caused by off-target covalent modification. Therefore, some non-covalent and non-peptoid protease inhibitors were also proposed. For example, Zhang et al. designed a series of highly active non-covalent, non-peptoid Mpro inhibitors based on the structure of perampanel [65]. Su et al. confirmed that Baicalin and its analogue Baicalein can non-covalently bind to the active site of Mpro, exhibiting potent antiviral activities in a cell-based assay 64]. Ratia et al. discovered GRL0617, a non-covalent inhibitor of SARS-CoV PLpro, in 2008 and proved to be effective in inhibiting SARS-CoV-2 infection [77,78]. Subsequently, Shen et al. designed a series of new PLpro inhibitors based on the scaffold of GRL0617 [58]. These non-covalent drugs may become new therapies for the treatment of SARS-CoV-2 infection. By tracing the sources of these active inhibitors, it is found that some of them are from the early virtual screening studies [65,79]. However, besides these known inhibitors, there are much more recommended compounds obtained from CADD studies for the further study. Our analysis showed that these CADD compounds exhibit high chemical spatial diversity, which is particularly valuable to guide the development of new drugs.

S, ACE2, and RdRp are also potential targets and were investigated by some CADD studies. Since the interface between S and ACE2 lacks typical pockets or cavities, which are generally required for small-molecule inhibitors, the docking studies on these two targets involve the selection of binding pockets. For S, all the existing reports aimed at targeting the receptor-binding domain (RBD), but focused on different binding sites. For example, Choudhary et al. [80], Sinha et al. [81], Alexpandi et al. [82], and Kalhor et al. [83] directly selected the contact surface that binding to ACE2 as the docking site and Mahdian et al. [38], Feng et al. [84] and Wei et al. [85] selected pockets near the contact surface for docking. For ACE2, the docking studies targeted the contact surface that binds to S RBD [38,80,86,87] or the catalytic pocket to block the enzyme activity and stabilize the closed conformation of ACE2, thereby shifting the relative positions of the receptor’s critical exterior residues recognized by SARS-CoV-2 [59,88,89]. However, no experimental evidence has shown the effectiveness of these strategies. To date, the promising drug leads targeting S or ACE2 to block SARS-CoV-2 infection are peptide binder and neutralizing antibodies (such as soluble human ACE2) [90,91,92,93,94]. For RdRp, the reported effective inhibitors are mainly nucleotide analog viral polymerase inhibitors, such as Remdesivir and Favipiravir, but the clinical outcomes are unsatisfactory. For example, several randomized clinical trials that investigated the efficacy of Remdesivir for COVID-19 treatment produced inconsistent results [95,96,97,98]. Favipiravir, compared with Arbidol, did not significantly improve the clinical recovery rate at day 7, but it did significantly improve the latency to cough relief and decreased the duration of pyrexia [99]. The meta-analysis conducted by Shrestha et al. concluded that patients had clinical and radiological improvements following the treatment with Favipiravir in comparison to that of the standard care though no significant differences on viral clearance, oxygen support requirement, or side effect profile [100]. One possible reason is that the exoribonuclease of SARS-CoV-2 provides the proofreading capacity to RdRp thus exerts relatively high resistance of RdRp on nucleotide analog inhibitors [101,102]. These nucleotide analogs usually bind to the RdRp active site (NTP binding site) [103], which is also the main site for the development of RdRp inhibitors using CADD methods [104]. Recently, the cryo-electron microscopy structure of the viral RdRp bound to suramin, a non-nucleotide inhibitor, reveals two new promising binding sites, which are the binding sites of the RNA template strand and primer strand [105]. The new structural mechanism is conducive to CADD studies.

## 5. Conclusions

In summary, starting from the structures of potential COVID-19 drug targets in complex with known inhibitors, our work first systematically explored the potential binding sites and binding modes to gain insights for drug discovery by CADD. We then collected and analyzed the 779 recommended compounds from 181 CADD studies based on the structures of target proteins, aiming to identify novel inhibitors from them. All data can be retrieved from our database, DrugDevCovid19, with a user-friendly interface at http://clab.labshare.cn/covid/php/index.php (accessed on 13 November 2020). Through comparison studies between CADD compounds and co-crystallized inhibitors, the 779 CADD compounds showed diverse chemical scaffolds compared with the co-crystallized inhibitors, meanwhile, covering the variants of co-crystallized inhibitors. Nineteen recommended compounds with highly similar Mpro inhibitors are worth further study, including perampanel (an anti-epileptic drug), 14 flavonoid analogues with a similar skeleton structure to Baicalein and Myricetin, and four peptoids with similar docking modes as those of co-crystallized inhibitors.

## Figures and Tables

**Figure 1 molecules-27-00683-f001:**
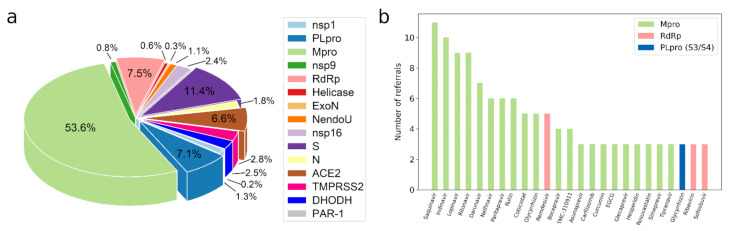
The statistics of compound records retrieved from CADD studies. (**a**) The distribution of compound record numbers classified by drug targets. (**b**) The CADD recommended compounds reported in more than two publications. Their targets included Mpro, RdRp, and PLpro. S3 and S4 are the two binding sites of PLpro.

**Figure 2 molecules-27-00683-f002:**
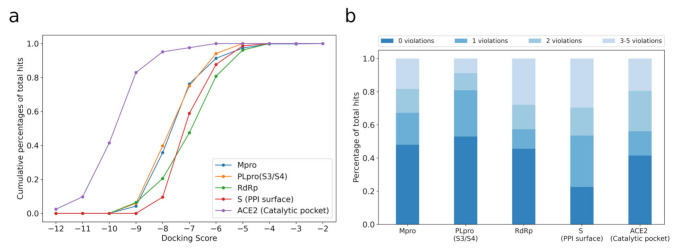
The distribution of docking scores and drug-like properties of CADD recommended compounds. (**a**) The cumulative percentage of CADD recommended compounds with docking scores increasing. (**b**) The percentage of CADD compounds following the Lipinski’s rules. Zero violations indicates the compounds that follow the Lipinski’s “Rule of Five”: MW ≤ 600, HBA ≤ 10, HBD ≤ 5, NrotB ≤ 10, and −2 ≤ cLogP ≤ 5; 1 and 2 violations indicate the compounds that violate one and two of the above five rules, respectively; 3–5 violations indicate the compounds that violate at least three of the five rules.

**Figure 3 molecules-27-00683-f003:**
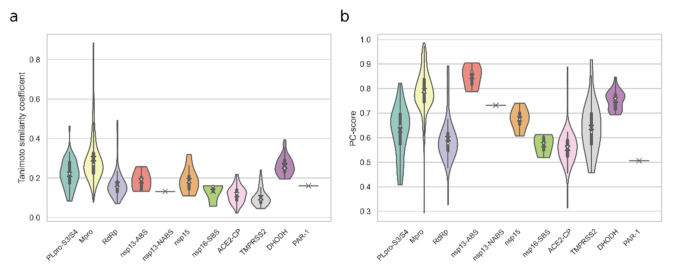
The violin plots of molecular similarities between the CADD recommended compounds and known inhibitors for each drug target. (**a**) The distribution of Tanimoto similarity coefficients between the CADD recommended compounds and known inhibitors. (**b**) The distribution of PC-score between the CADD recommended compounds and known inhibitors. The targets and binding sites include PLpro S3/S4 pocket (PLpro-S3/S4), catalytic pocket of Mpro, RdRp, nsp15, ACE2 (ACE2-CP), TMPRSS2 and DHODH, the ATP binding site (nsp13-ABS) and nucleic acids binding site (nsp13-NABS) of nsp13, the SAM binding site of nsp16 (nsp16-SBS), and the active site of PAR-1. The position of ‘X’ and the white points represent the average values and median values respectively and the violin range is limited within the range of the observed data.

**Figure 4 molecules-27-00683-f004:**
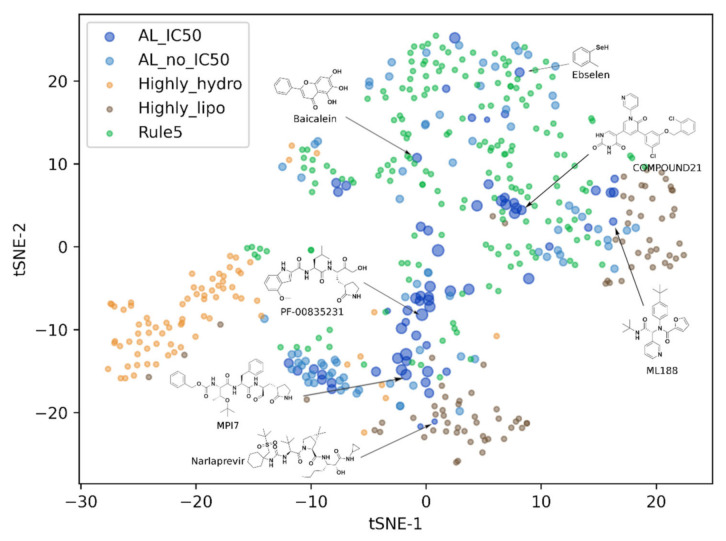
The t-distributed stochastic neighbor embedding (tSNE) map of the chemical spatial features including cLogP (calculated lipophilicity), HBA (hydrogen-bond acceptor), HBD (hydrogen-bond donor) and MW (molecular weight) for each CADD recommended compounds and the co-crystallized inhibitors against Mpro. AL_IC50 represents the inhibitors with known IC50. The size of the circles refers to the IC50 of the compounds (the larger size indicates the smaller IC50). AL_no_IC50 represents the inhibitors with unknown activity data; Rule5 represents the CADD compounds that meet the criteria of MW ≤ 600, HBA ≤ 10, HBD ≤ 5, and −2 ≤ cLogP ≤ 5; Highly_lipo refers to the CADD recommended compounds that violate one or more of the above criteria and cLogp > 2 (i.e., compounds with high lipophilicity); Highly_hydro refers to the CADD recommended compounds that violate one or more of the above criteria and cLogp < 2 (i.e., compounds with high hydrophilicity).

**Figure 5 molecules-27-00683-f005:**
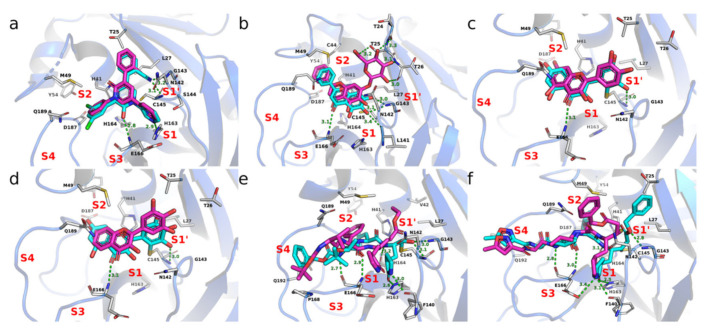
The binding modes of CADD recommended compounds of Mpro compared with the corresponding co-crystallized structures. The CADD recommended compounds are colored in magenta, including perampanel (**a**), Baicalin (**b**), quercetin (**c**), robinetin (**d**), DB04692 (**e**) and compound-23727975 (**f**). The co-crystallized inhibitors are colored in cyan, including COMPOUND4 (**a**), Baicalein (**b**), Myricetin (**c**,**d**), MPI6 (**e**), and N3 (**f**).

**Figure 6 molecules-27-00683-f006:**
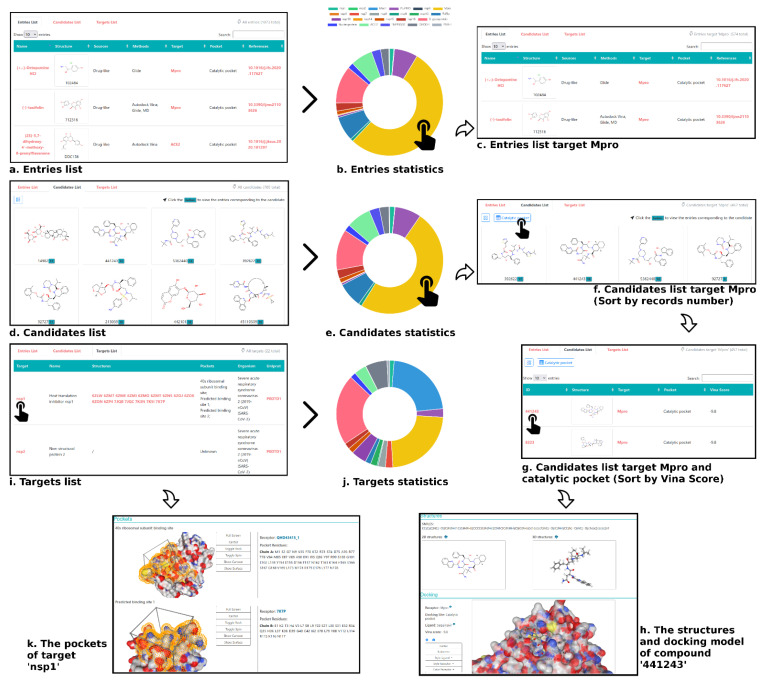
The web interface of DrugDevCovid19 server. (**a**) The ‘Entries List’ presented all the compound records retrieved from literature. (**b**) The donut chart displayed the statistics of compound records classified by the drug target. (**c**) The ‘Entries List’ presented compound records targeting Mpro. (**d)** The ‘Candidates List’ presented all the non-redundant compounds contained in the records. (**e**) The donut chart displayed the statistics of non-redundant compounds classified by the drug target. (**f**) The ‘Candidates List’ presented the non-redundant compounds targeting Mpro (sorted by record number). (**g**) The ‘Candidates List’ presented the non-redundant compounds targeting catalytic pocket of Mpro (sorted by Vina score). (**h**) The structural information and docking conformation of compound 441243 (PubChem CID) targeting the catalytic pocket of Mpro. (**i**) The ‘Targets List’ presented all the potential drug targets. (**j**) The donut chart displayed the statistics of experimentally solved structures classified by the drug target. (**k**) The 3D structural features and surface residues of the active sites of nsp1.

**Table 1 molecules-27-00683-t001:** The main drug targets for the treatment of SARS-CoV-2 used in CADD studies.

Targets	Short Name	Organism	Solved Structures (Solved Complexes)
Host translation inhibitor nsp1	nsp1	SARS-CoV-2	21 (0)
Non-structural protein 3	nsp3, PLpro	SARS-CoV-2	308 (262)
3C-like proteinase	nsp5, 3CLpro, Mpro	SARS-CoV-2	369 (177)
Non-structural protein 9	nsp9	SARS-CoV-2	12 (0)
RNA-directed RNA polymerase	nsp12, RdRp	SARS-CoV-2	28 (9)
Helicase	nsp13, Hel	SARS-CoV-2	67 (58)
Proofreading exoribonuclease	nsp14, ExoN	SARS-CoV-2	11 (0)
Uridylate-specific endoribonuclease	nsp15	SARS-CoV-2	44 (26)
2’-O-methyltransferase	nsp16	SARS-CoV-2	28 (24)
Spike glycoprotein	S	SARS-CoV-2	466 (13)
Nucleoprotein	N	SARS-CoV-2	21 (0)
Angiotensin-converting enzyme 2	ACE2	*Homo sapiens*	58 (1)
Transmembrane protease serine 2	TMPRSS2	*Homo sapiens*	1 (1)
Dihydroorotate dehydrogenase	DHODH	*Homo sapiens*	79 (75)
Proteinase-activated receptor 1	PAR-1	*Homo sapiens*	5 (1)

Note: Solved structures refer to the number of solved 3D structures in the PDB; solved complexes refer to the number of solved 3D structures complexed with ligands (binding in the active sites) in the PDB.

**Table 2 molecules-27-00683-t002:** Statistics of studies on the SARS-CoV-2 inhibitor discovery using CADD.

Target	#Refs	#Hits	Target	#Refs	Hits
nsp1	2	14	nsp16	2	25
PLpro	12	74	S	34	114
Mpro	124	457	N	5	18
nsp9	2	9	ACE2	16	69
RdRp	23	70	TMPRSS2	6	29
Helicase	2	5	DHODH	1	27
ExoN	2	2	PAR-1	1	2
NendoU	2	12			

## Data Availability

Not applicable.

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
