# Peer review of "DrugDevCovid19: An Atlas of Anti-COVID-19 Compounds Derived by Computer-Aided Drug Design"

_molecules, 2022, doi:10.3390/molecules27030683_

Round 1
Reviewer 1 Report
The manuscript of Yang Liu et al. presents an important piece of research for the computational drug design and discovery and pharmacology community aiming to develop COVID-19 therapeutics. The authors are using an extensive computational platform including docking experiments (mainly by AutoDock Vina among others), ADMET predictions and advanced statistical tools to integrate the main chemical and pharmaceutical agents used to produce co-crystal structures with the main pharmacological protein targets of interest for the COVID-19 therapeutics. Specifically, the authors reviewed and analyzed 779 compounds against 16 target proteins from 181 CADD publications.
Using the structures of potential COVID-19 drug targets in complex with known inhibitors, the authors are systematically exploring the potential binding sites and binding and conformational modes to gain further insights for drug discovery by CADD. Further, the authors are then collecting
and analyzing the recommended compounds from the current CADD studies based on the structures of target proteins, aiming to identify novel inhibitors from them.
Remarkably, the authors organized all the data and constructed a freely available database, DrugDevCovid19 (http://clab.labshare.cn/covid/php/index.php), to facilitate the development of COVID-19 therapeutics.
The paper is very well written for the computational theoretical chemistry section of MDPI, and I am recommending it for being accepted for publication with one minor revision.
I will still recommend the addition of one paragraph in the discussion section where the authors should comment on the off-target analysis for each of the analyzed compounds.
Specifically, the authors should argue if the new implemented website platform would allow to perform cross-docking experiments against potential off-target proteins from the human host, structurally related to the main drug targets for the treatment of SARS-CoV-2 used in CADD studies shown in Table-1.
The prediction of such off-targets in the human host would help to develop COVID-19 therapeutics with less side effects and reduced toxicity.
Reviewer 2 Report
Several such computer-aided drug design (CADD) studies have been already published. Given the ongoing COVID-19 pandemic, some aspects of these studies are useful. However, some key elements are missing in this paper.
Some points for improvement of the study:
Purely computational work usually requires some experimental evidence to support the study's findings. Authors should perform in-vitro and in-vivo studies to prove that some of their recommended compounds are worth as drugs. This will validate the effectiveness of the previous in-silico method as well as the author's recommendations.
Summary (Line 447-451): The author should make a separate section ‘Conclusion’ and summarize the results of the study instead of the methods.
Author Response
Response to Reviewer 2 Comments
We would like to thank the reviewer very much for taking the time and effort to review our paper so thoroughly. We have carefully considered the comments, which have helped us to make improvements to our work. Our point by point responses to the comments are presented below.
Point 1: Purely computational work usually requires some experimental evidence to support the study's findings. Authors should perform in-vitro and in-vivo studies to prove that some of their recommended compounds are worth as drugs. This will validate the effectiveness of the previous in-silico method as well as the author's recommendations.
Response 1: We thank the reviewer for the suggestion. Indeed in-vitro and in-vivo studies are the crucial step in drug discovery, and they are also expensive and time-consuming. The main purpose of this study is to statistically analyze the current CADD researches on anti-COVID-19 drug design and conduct comparison studies between them with the reported inhibitors. We discovered that some of the CADD compounds exhibit highly similar features with reported inhibitors in terms of binding modes, chemical properties, scaffold, drug likeness etc., while some others showed diverse chemical scaffolds with few drug likeness. And our work revealed some critical binding modes of known active inhibitors, which are critical to future drug design. Particularly, we summarized the one-year’s CADD studies and constructed the database, which is an easy-to-check CADD record of the anti-COVID-19 drug design. Nowadays, drug discovery is a highly complex work that needs concerted efforts of both theoretical and experimental researchers. Thus we argue that our computational analysis can also provide insights to the anti-COVID-19 studies.
Point 2: Summary (Line 447-451): The author should make a separate section ‘Conclusion’ and summarize the results of the study instead of the methods.
Response 2: We thank the reviewer for the suggestion and added a separate section ‘Conclusion’ to summarize the results of our study (Please see ‘Conclusion’ section).
Reviewer 3 Report
The manuscript submitted by Cao and co-workers deals with the publication of a new open server with a focus on the use of virtual screening whiting the current COVID-19 emergency.
This referee agrees with the main purpose of the present work, as computational chemistry and last molecular models might help in the design of an efficient antiviral drug. Vaccines are not enough, antiviral must be also discovered.
I have a long experience in both the use of theory to design new drugs and in their confirmation in real labs with in vitro and in vivo models. Unfortunately, docking score functions have not provide any real drug so far. Even if such functions are subsequently improved with higher levels of theory (MD, MMPBSA/MMGBSA, FEP...) most of drugs fails when tested in real virus test with human cells.
I am sorry to say that the reported data in that server do not add new insight into what is already known. Indeed, very similar severs are already only. See for instance the one created by Harvard University at that link:
https://vf4covid19.hms.harvard.edu
Please, notice I am not one of the authors and I am NOT asking for a citation. That server is used to illustrate that very similar solutions are already available (even larger).
In short:
- Virtual screening and docking have been largely applied for two years without success.
- Score must be always refined with more accurate levels of theory if real medical applications are looked for.
- Similar servers are available.
The paper is not suitable for Molecules. After reviewing all sister journals in MDPI, I feel it might fit to Compounds.
Author Response
Response to Reviewer 3 Comments
We would like to thank the reviewer very much for taking the time and effort to review our paper so thoroughly. We have carefully considered the comments, which have helped us to make improvements to our work. Our point by point responses to the comments are presented below.
Point 1: Virtual screening and docking have been largely applied for two years without success.
Response 1: We thank the reviewer for the comments. Currently approved anti-COVID-19 drugs (e.g. PF-07321332) or those in clinical phase (e.g. PF07304813, 13b) are derived from optimization of existing antiviral drugs, however, it is unfair to impute blame to CADD studies, because it is extremely difficult to obtain therapeutic drugs in a short term. Actually, some CADD derived candidates have shown promising progress. For example, Perampanel, an anti-epileptic drug, have been screened and validated by Gimeno et al. from the approved drug library[1-2]. Subsequently, Zhang et al. designed a series of inhibitors of Mpro with ca. 20 nM potency based on the structure of Perampanel[3]. Another example is Baicalin, which is a potential Mpro inhibitor screened from 40 antiviral phytochemicals using virtual screening[4], and related studies show that baicalin and its analog baicalein exhibited potent antiviral activities in a cell-based assay[5]. In addition, there are also some approved drugs such as Remdesivir, Chloroquine and Camostat that have early CADD studies to demonstrate their strong binding ability with the corresponding targets[6-8]. Therefore, it is of value to summarize the current CADD discoveries, and find out the strong and weakness of current works in order to further improve relevant studies.
- Gimeno A, Mestres-Truyol J, Ojeda-Montes MJ, et al. Prediction of novel inhibitors of the main protease (M-pro) of SARS-CoV-2 through consensus docking and drug reposition. Int. J. Mol. Sci. 2020; 21:3793
- Ghahremanpour MM, Tirado-Rives J, Deshmukh M, et al. Identification of 14 Known Drugs as Inhibitors of the Main Protease of SARS-CoV-2. ACS Med. Chem. Lett. 2020; 11:2526–2533
- Zhang CH, Stone EA, Deshmukh M, et al. Potent Noncovalent Inhibitors of the Main Protease of SARS-CoV-2 from Molecular Sculpting of the Drug Perampanel Guided by Free Energy Perturbation Calculations. ACS Cent. Sci. 2021; 7:467–475
- Islam R, Parves MR, Paul AS, et al. A molecular modeling approach to identify effective antiviral phytochemicals against the main protease of SARS-CoV-2. J. Biomol. Struct. Dyn. 2020; 1–12
- Su H xia, Yao S, Zhao W feng, et al. Anti-SARS-CoV-2 activities in vitro of Shuanghuanglian preparations and bioactive ingredients. Acta Pharmacol. Sin. 2020; 41:1167–1177
- Chang Y, Tung Y, Lee K, et al. Potential therapeutic agents for COVID-19 based on the analysis of protease and RNA polymerase docking. Preprints 2020;
- Abdulfatai U, Uzairu A, Shallangwa GA, et al. Molecular Docking Analysis of Chloroquine and Hydroxychloroquine and Design of Anti-SARS-CoV2 Protease Inhibitor. Mod. Appl. Sci. 2020; 14:52
- Escalante DE, Ferguson DM. Structural modeling and analysis of the SARS-CoV-2 cell entry inhibitor camostat bound to the trypsin-like protease TMPRSS2. Med. Chem. Res. 2021; 30:399–409
Point 2: Score must be always refined with more accurate levels of theory if real medical applications are looked for.
Response 2: We agree with the reviewer that the current docking-based scoring functions have some drawbacks. However, higher levels of theories (MD, MMPBSA/MMGBSA, FEP...) are computationally expensive, but gather little gain in docking screening[9]. In contrast, comparative structure–activity relationship analysis can bring more valuable information. We have some successfully stories combining the docking screening and comparative analysis of structure–activity relationship to discovery drug candidates. For example, we found an old drug Nebivolol which can be repurposed to bind with FOXO3 and degrade EGFR in non-small cell lung cancer cells. This work has been published in Nature Communications[10] recently. Therefore, we combined the docking and comparative analysis of CADD candidates with the reported co-crystal inhibitors. We discovered that some of the CADD compounds exhibit highly similar features with reported inhibitors in terms of binding modes, chemical properties, scaffold, drug likeness etc., while some others showed diverse chemical scaffolds with few drug likeness. And our work revealed some critical binding modes of known active inhibitors, which are critical to future drug design. Particularly, we summarized the one-year’s CADD studies and constructed the database, which is an easy-to-check CADD record of the anti-COVID-19 drug design.
- Åšledź P, Caflisch A. Protein structure-based drug design: from docking to molecular dynamics. Curr. Opin. Struct. Biol. 2018; 48:93–102
- Niu M, Xu J, Liu Y, et al. FBXL2 counteracts Grp94 to destabilize EGFR and inhibit EGFR-driven NSCLC growth. Nat. Commun. 2021; 12:5919
Point 3: Similar servers are available.
Response 3: We have checked the VirtualFlow@Covid19 server (https://vf4covid19.hms.harvard.edu) mentioned by the reviewer. In general, VirtualFlow@Covid19 and ours are about the virtual screening studies but there are essential distinctions between the two servers.
- Firstly, our database is a collection of numerous CADD studies, bringing together candidates selected by different researchers based on diverse computational methods and their experiences. It is used for analyzing and evaluating the current CADD drug designs. In contrast, VirtualFlow@Covid19 is an ultra-large virtual screening platform, which is aiming to recommend possible drug candidates by their own computation pipeline. The two servers are inconsistent with their goals and methods.
- Secondly, our server provides abundant information about the CADD candidates including structures, origin, screening methods, reference, binding site residues, docking score and pose, physicochemical properties, relevant off-target information and pharmacodynamics if available. Each of the compounds is illustrated in the 3D view of the binding mode, aiming to help the users analyze the drug-likeness of the recommended compounds and their binding ability to the target. In contrast, VirtualFlow@Covid19 only offers download links of the top 1 million hits list of each target without docking poses with options for users to select the hits by physicochemical properties. Hence the contents of the two servers are different.
- Thirdly, our server integrated some online tools for more professional analysis by the user, such as the NGL viewer that can show the molecular interactions, off-target analysis based on homologus proteins or similar ligands, etc. In contrast, VirtualFlow@Covid19 only provide hit compound files without any further analysis tools. Therefore, the two servers are functionally different.
In all, VirtualFlow@Covid19 and our databases are not similar. And we also did a survey on the published databases on drug discoveries about COVID19, such as CORDITE (https://cordite.mathematik.uni-marburg.de/#/). Compared to those works, our database is unique in data collection and the interactive interface for data analysis. Therefore, we believe our work can provide insights to the community of anti-COVID19 drug discovery.
Round 2
Reviewer 2 Report
I am satisfactory with authors response and recommend the revised manuscript for publication in the journal Molecules.
Reviewer 3 Report
The paper must be largely improved to have any impact in the field of drug repurposing with a focus on COVID-19.
It is not my goal to slow publication down. As Referees 1 and 2 seems to be less skeptic than me, and once the authors replied to my comments by solely including more references, I have no further comments/suggestions to add.
However, I am still positive the paper better fits to less demanding journal rather than to Molecules.